# God and the Problem of Evil: An Attempt at Reframing the Debate

Brett Wilmot

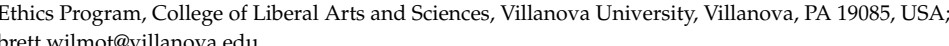

Ethics Program, College of Liberal Arts and Sciences, Villanova University, Villanova, PA 19085, USA;
brett.wilmot@villanova.edu

**Abstract:** This article attempts to reframe the traditional account of the problem of evil for God's existence. The philosophical debates about the problem of evil for the existence of God within the traditional framework do not exhaust the available options for conceiving of God's perfection, including our understanding of God's power and God's relationship to the world. In responding to the problem of evil, rational theists should seek a reformulation of divine perfection consistent with God's existence as both necessary and as morally relevant to human life in a manner that does not collapse in the face of the problem of evil. The neoclassical account of God's nature as developed in the tradition of process philosophy is presented as an alternative that meets these requirements.

**Keywords:** process philosophy; theism; ontological argument; theodicy; problem of evil; metaphysics

## 1. A Little Background

> J.B.: "If God is good, he is not God; if God is God, he is not good; take the even, take the odd." (MacLeish 1958)

> Anselm: "God is that, than which nothing greater can be conceived."

James Sterba's (2019) recent book revisits the perennial theodicy debate within traditional theism and responds to a range of contemporary efforts to defend the logic of God's existence in the face of the presence of horrendous evil in the world (both moral and non-moral). At the heart of this debate is the question of whether our experience of evil in the world counts against the existence of God understood as all powerful, all knowing, and perfectly good. If God is all powerful and all knowing, then God must not be perfectly good to allow horrendous evil, or if God is perfectly good, then God's power must be limited given that such evils occur. The argument suggests that attentiveness to the many horrendous moral and natural evils we find in the world cannot help but undermine belief in the God of traditional theism, in whom the virtues of omnipotence, omniscience, and moral goodness are thought to coincide. Take the even, take the odd.

I am generally sympathetic with Sterba's position with regard to the vulnerabilities of traditional theism to the problem of evil. My goal in what follows is to suggest an alternative account of divine perfection that is invulnerable to the line of critique Sterba advances in his book. Specifically, I want to advocate for a version of "neoclassical theism" in the vein of Charles Hartshorne's process philosophy as one such alternative that I believe avoids the pitfalls of the problem of evil while providing a compelling account of God's perfection, including God's necessary existence and relevance for the moral lives of human beings. This essay is an attempt to present and clarify how that approach to God's existence avoids the problem of evil and justifies our continued affirmation of the existence of God.

## 2. The Aim of the Essay

I should be clear on a few points here at the outset. As indicated above, I am generally sympathetic with Sterba's challenge to traditional theism in light of the problem of evil. I, too, find it difficult to reconcile a traditional understanding of God's perfection with the

degree of moral and natural evil the world, and he has advanced a particularly robust account of this problem. The argument I will develop in this essay takes up a different challenge: Is there an alternative way of understanding God's perfection such that the problem of evil no longer serves as a logical objection to the existence of God? I believe that there is, but the limits of this article will require me to be more suggestive than systematic in presenting this alternative. Nonetheless, I hope the view advanced here will be attractive in no small part because it presents a path for securing God's status as an unsurpassable individual and the proper object of our ultimate concern in a manner that renders the problem of evil irrelevant when defending the existence of God.

While the approach for which I advocate does not take up in great detail the specific points made by Sterba in his critique of traditional theism, there are some parallels worth noting. While the position I will advance is not a version of skeptical theism in the vein of Michael Bergmann's in chapter 5 of Sterba's (2019) book, I will suggest a way in which epistemic humility on our part might still be relevant to this topic when evaluating God's choices about the initial conditions governing a particular cosmic epoch (the natural laws of the current universe that provide the framework for coordinating the activities of finite individuals). I will also argue that God is not a moral agent and that the ascription of moral goodness as part of God's perfection is a category mistake. My development of this point, however, will not rely on the line of argument for that view advanced by Brian Davies in chapter 6 of the book (Sterba 2019). Instead, I will argue that God stands in an asymmetrical relationship with moral goodness: God experiences and values the moral goodness of finite beings as a species of goodness more generally, but properly understood, God does not exercise moral agency even as God's activity is value maximizing by necessity. It is simply the nature of divine activity to act on, and respond to, the world in a manner that optimizes value for future purpose, where part of that value includes the distinctive contributions made by moral agents. Moral goodness, on this account, should be restricted in reference to the free choices that finite rational individuals make with respect to maximizing value for future purpose, where it is always possible for such individuals to choose a lesser value.

Acts of moral goodness contribute to the divine good, but God's agency is dissimilar to our own in ways that render moral choice meaningless in the divine context. God's activity minimizes evil as a corollary of God's value maximizing, but if we understand moral agency as operating in the space of the freedom to choose between greater and lesser value (where moral evil involves the choice of a lesser value), then God does not exercise moral agency (even perfect moral agency in the sense of possessing a "holy will" per Kant, where God obeys the moral law by necessity). Because God's actions are value maximizing as a metaphysical necessity, it is simply a category mistake to attribute the conditions of moral agency to God, from which it follows that an account of the divine nature should not include the attribute "morally perfect." As counterintuitive as this might sound, particularly in the context of traditional theism, I believe this turn in philosophical theology has significant benefits, not the least of which is to remove the threat of the problem of evil to the logic of God's existence.

My aim in this essay is not to provide a full expression and defense of neoclassical theism. Rather, I want to suggest that the standard framing of the problem of evil unfolds within a particular set of assumptions about divine perfection, God's relationship to world, and how power is shared in the context of those relationships. These are traditional assumptions that reflect the dominant discourse within orthodox theism in the Abrahamic context. In advancing an alternative approach, I recognize that it will take us beyond that framework in ways that will be viewed as heterodox by most traditional theists. Still, I think it is worth recognizing that philosophical debates about the problem of evil for the existence of God within that orthodox framework do not exhaust the available options for conceiving of God's perfection, including our understanding of God's power and God's relationship to the world. In philosophical discourse, heresy should not be an objection to considering possible alternatives. There may remain options available to traditional theists in pushing back against the arguments advanced by Sterba, but my own view is that his

objections to recent apologetic efforts are persuasive. That said, where Sterba concludes from his achievement that God's existence is logically impossible, I am inclined to counter that what he has shown is the need for a better formulation of divine perfection consistent with God's existence as both necessary and as morally relevant to human life in a manner that does not collapse in the face of the problem of evil. To my mind, neoclassical theism provides such an alternative.

### 3. God's Necessary Existence

The problem of evil operates under the presumption that contingent, empirical matters are relevant to inferential judgments about the logical possibility of God's existence. According to Charles Hartshorne, however, this argument involves a conceptual error. "As Aristotle had seen, 'with eternal things to be and to be possible are the same.' If then the eternal God is not, the eternal God is impossible and could not have existed. But empirical arguments are addressed to contingent matters, what could be, but perhaps not, the case" (Hartshorne 1983, p. 58). With regard to the eternal, then, "empirical evidence is irrelevant" (Hartshorne 1983, p. 59). This is the point Hartshorne makes in discussing the ontological argument in the context of Hume: "[Hume] grants that its validity would dispose of the argument against theism based on the evils of the world" (Hartshorne [1965] 1991, p. 201). The greatest challenge to a coherent theism, it seems to me, is not the problem of evil but rather the problem of God's status as existing necessarily, that is, in some respect, as an eternal individual. The problem of evil emerges as a result of deficient understandings of God's nature and perfection. The solution, it seems to me, is not to develop increasingly sophisticated rejoinders to the problem of evil in defense of God's perfect moral agency but to reframe our understanding of God's nature and perfection such that it becomes clear that the existence of evil, even horrendous evil, simply is not relevant to determining whether God exists.

The more interesting challenge to a coherent theism, then, involves whether a persuasive account of God's perfection can be formulated that sustains God's necessary existence while providing clarification in terms of how God relates to the world that avoids the problem of evil altogether. Philosophically inclined theists should focus on that task rather than pursuing apologetics in response to the problem of evil. For the moment, there is broad skepticism regarding the possibility of engaging in the kind of metaphysical efforts associated with this pursuit, including the transcendental method of process philosophy. Still, there are times when the dominant consensus is wrong, and I suspect this is one of them. If there is hope of success in such a project, then I think that Hartshorne's method points us in the most likely direction of success.[1]

### 4. Neoclassical Theism and Divine Perfection: A Heterodox Alternative

On the approach that I am recommending, the role of the problem of evil in philosophical theology shifts. Rather than presenting an objection to the logic of God's existence, it serves merely to reveal conceptual error in a particular conception of the divine nature. The focus of the rational theist, then, should not be responding to the problem of evil by seeking to reconcile God's omnipotence, omniscience, and perfect moral agency with the amount of evil in the world. Instead, on the discovery that an account of divine perfection errs in locating God within the class of contingent beings, the rational theist should revisit her understanding of the divine nature in order to secure God's necessary existence on purely a priori grounds, that is to say, transcendentally.

In an email from 5 December 2016, Sterba suggested that my locating God within this modal category of necessary existence results, as in the case with Aristotle's First Mover, in a deity whose existence would be compatible with any degree of moral and natural evil in the world. Such a conception appears incompatible with the commitments of traditional

---

[1] I am inclined to think that Franklin Gamwell's development of Hartshorne's method addresses a number of potential problems, but the transcendental method employed by both is essentially the same. See, esp., Gamwell (2020).

theism and is one that Sterba suggested we should have little interest in defending because such a God would be irrelevant to our moral lives. At the very least, as Enlightenment deists recognized, while such a God may provide some explanatory benefits for thinking about the origins of the world, this God makes little contribution to our moral lives on an ongoing basis. The traditional account of God's moral perfection implies that God acts in or on the world in a morally significant manner, and it is God's moral status that grounds, in part, our attention to God in our own moral reasoning. On reflection, I believe Sterba is right, in part, in his judgment here. It is an error to characterize God as both eternal and morally perfect, as exercising agency that is in some sense subject to the general form of moral evaluation proper to finite rational beings like ourselves. The proper response, I want to suggest, is to resolve the horns of this dilemma by rejecting God's moral agency while preserving God's status as eternal. God should not be included in the class of moral agents, and as a result, we should avoid ascribing moral perfection to God. I will expand on these points later in the essay.

It is the second of Sterba's judgments that I want to challenge here: that a God conceived to be eternal is not one that we should have an interest in defending because this God is irrelevant to our moral lives. To this end, I will endeavor to show that the God of neoclassical theism plays a very different role than Aristotle's First Mover (or the deists' God), such that God's consequent nature as a universal subject preserves God's status as our final end (the comprehensive *telos* to which our activities make their ultimate contributions). In recognizing that God is not properly characterized as a moral agent or as possessing the property of moral perfection, we need not conclude that God is indifferent or irrelevant to moral goodness or that God does not do all that God properly can do to maximize value (both moral and non-moral). Unlike Aristotle's God, who initiates motion in the universe but who provides no comprehensive *telos* for the activity that follows, the God of neoclassical theism provides the cosmic purpose in relation to which all value has its final significance. This, then, the neoclassical conception of God shares with classical theism: God is both *alpha* and *omega*, first and last. God sets the initial conditions under which finite individuals realize value, and God's subjective experience is that to which all value (moral and non-moral) makes its ultimate contribution. It is in this context that we can say that God has an asymmetrical relationship with moral goodness. Moral goodness makes a distinctive contribution to the divine experience, even if God does not exercise moral agency or produce moral goodness in the exercise of God's power.

How might we go about reframing our understanding of God's nature such that God's not being a moral agent or morally perfect does not count against God's perfection and unique status in the cosmic economy of value? To achieve this end, I believe that we need to reconsider the property of omnipotence. Rather than start with the classical intuition that God's perfection implies omnipotence (literally, the possession of all power), we might reframe our approach around the idea of God's greatness in contrast to other individuals, where the relevant meaning of perfection is that God's power is unrivaled (insuperable), not that God is omnipotent. According to Hartshorne, "'Greatness' means having whatever properties it is better to have than not to have, as compared to other conceivable individuals" (Hartshorne [1965] 1991, p. 202). A better way to express God's special status as an individual, Hartshorne suggests, is to speak of God's being "unsurpassable" in contrast to other non-divine individuals. When considering God's perfection, we should keep in mind that "it may very well not be 'best' to be 'omnipotent', in the sense which generates the problem of evil in its classical form" (Hartshorne [1965] 1991, p. 202). In what follows, I want to suggest some ways in which we might reconsider God's greatness and relationship to the world using the framework of neoclassical theism that avoids the pitfalls of that God's omnipotence generates for the traditional account of the divine nature.

## 5. God and the World: Co-Eternal

On the traditional account of the divine nature, moral perfection is a quality attributed to God. There are two primary reasons for this. First, on the traditional account, God's

perfection is defined in terms of the possession of all positive qualities to the highest degree. If moral perfection is a positive quality possessed by any individual, then it must be found preeminently in God. Second, and related to the first, is the understanding of God's role in the creation of the world. This is the doctrine of creation ex nihilo, where the existence of all things outside of God are the result of God's creative activity, where each thing exists only insofar as it possesses imperfectly some of the properties perfectly realized in the divine nature. On the traditional account of causal efficacy, an effect is found preeminently in the cause. If God is the original cause of everything that exists, then whatever qualities and powers we find in the creation are but imperfect reflections of the fullness of those qualities and powers in God. Since we are created by God, our moral powers, imperfect as they are, must reside in their fullest sense in God, and this is what opens the door to the problem of evil as a challenge to the logic of God's existence. God's omnipotence and moral perfection are set on a collision course given our sense that God could and should have prevented the horrendous evil we find in the world. So, either God lacks sufficient power to prevent such evil, or God is not morally perfect because of the evil God permits; the absence of either attribute is sufficient to conclude that God, as traditionally conceived, does not exist. Take the even, take the odd.

So, how might a neoclassical conception of divine greatness reframe our understanding of God's perfection, including God's relationship to the class of contingent, non-divine individuals, that can avoid the horns of this dilemma? In the place of omnipotence and creation ex nihilo, the neoclassical approach asserts that the class of all continent individuals is co-eternal with God (clearly not an orthodox view). Creation ex nihilo is not obviously preferable, logically speaking, to holding that the class of contingent individuals is co-eternal with God, though it involves significantly modifying our understanding of God's creative activity and the scope of God's power in relation to the world. Per Hartshorne, "greatness" means having whatever properties it is better to have than not to have, as compared to other conceivable individuals. As we will see, there is an enormous difference between God's eternal existence as a necessary individual and the necessary existence of a class of finite individuals, each of which on its own exists contingently. While creation ex nihilo is assumed within orthodox theism as part of its account of divine perfection and its understanding of causation, that by itself does not show its preferability in terms of the conceivable options, particularly if the traditional account gives rise to the problem of evil. Again, heresy is no objection within philosophical theology.

Following the neoclassical approach, God is the sole necessary individual, and the set of contingent individuals is never empty: there is always a contingent world to which God relates as a universal subject. According to Alfred North Whitehead, "the final real things of which the world is made up" are microscopic actualities or actual entities, each of which decides how to unify the past in order to serve the future (Whitehead [1929] 1978). God, too, is an actual entity, but as we will see, God's decisions about unifying the past in order to serve the future are value maximizing by necessity, in contrast to the decisions of moral agents, who are capable of choosing between greater and lesser value. To clarify the point further, God's choices are always among possibilities that maximize value equally, such that the choice among these possibilities is non-moral. Rather than omnipotent, God has all the power any one individual could have but not all the power there is, given that finite individuals also possess powers proper to their nature, powers that are not simply imperfect iterations of divine powers or subject to divine fiat. As we will see, God exerts cosmic influence, and it is in God's subjective experience that all value finds its ultimate significance. Still, there is also real, non-trivial power in the set of finite actual entities that make up the world.

While it is no longer appropriate to speak of God as omnipotent, it is still the case that God's power is "unsurpassable" by any member of the class of finite individuals. To be sure, there is a great deal at stake in this reformulation, not the least of which that it forces us to rethink the fundamental relationship between cause and effect found in the classical account (where an effect exists preeminently in its cause). More relevant for our

purposes, this shift opens space for reframing our understanding of God's relationship with states of affairs in the world, including the existence of horrendous evil. It remains, however, to flesh out the conception of God's perfection, or greatness, that follows from this metaphysical perspective (God as the sole necessarily existing individual in relation to a necessary class of contingent individuals, the set of which is never null, because "nothing exists" is impossible).[2] How might we express the idea of God's perfection under these new conditions, such that it remains proper to describe God's power as unsurpassable, or unrivaled, with respect to other individuals but that also avoids the problem of evil as an objection to God's existence?

On the neoclassical account, God interacts with the world as a whole and is affected by the world in all of its particularity. God exercises a kind of sovereign influence on the world, but this influence is constrained by the real power and freedom of finite actual entities, power and freedom that is not simply derivative of God's power through the traditional account of creation ex nihilo. Finite individuals have powers proper to their existential status, powers that are not simply imperfect manifestations of qualities found perfectly in the divine nature. One way to express God's perfection in this context involves understanding God's activity as necessarily value maximizing within the scope of God's power to influence non-divine activity in the world. This should not be understood as an exercise of divine will where God faces better and worse options but obeys the moral law by necessity (Kant's idea of a "holy will"). Instead, as a transcendental principle, divine activity necessarily maximizes the value possible as a result of the past actions of finite individuals through God's decision for future purpose. All finite individuals exercise real, non-trivial power of their own in making decisions about value for future purpose, and moral agents do so as well but with this difference. Finite rational individuals confront the possibility of choosing a lesser value in their decisions for future purpose. Because moral agents possess real freedom and power to act contrary to the divine purpose, evil, even horrendous evil, is always possible given the existence of finite rational individuals (moral agents). All existing individuals decide their contribution to future value, moral agents must choose between greater and lesser value when making this decision, and God alone acts in relation to the whole with an aim for the future that is value maximizing by necessity.[3]

Framed this way, the evil that exists in the world is irrelevant to the question of God's existence. Moral evil in particular is a potential feature of any reality in which there exist individuals with the capacity for choosing between greater and lesser value (moral agents), and moral evil is always the result of the exercise of finite freedom in spite of God's influence to the contrary. God's power to influence comprehensively is unsurpassed by our own powers of finite influence, but God's power is not absolute, since non-trivial power always exists in the members of any set of finite individuals. This difference allows space for finite individuals to contribute value to the divine life (the contribution of real novelty as a result of the exercise of finite freedom and power—something achieved by all actual

---

[2]  I suspect that for many, the contingency of all members of the set of finite individuals suggests the contingency of the set as a whole. But there's no reason to infer this conclusion about the set based on the contingency of its members. The possibility of there being nothing at all relies on an inference from our ability to conceive of the non-existence of any particular to the possibility of conceiving of nothing at all. While "nothing exists" may appear to name a conceivable alternative to "something exists," it is worth noting that it is impossible to distinguish between "nothing at all" and the strictly inconceivable. For example, a contradiction such as a "round square" literally identifies nothing at all: it is a putative thought with no object. It follows that "nothing at all" cannot be distinguished from a contradiction, and the inconceivable cannot serve as a possible alternative to "something exists." The conclusion follows that "something exists" is logically necessary, which is precisely what is meant by stating that the set of finite individuals is never null in spite of the contingent status of all of its members.

[3]  Franklin Gamwell suggests another way to make this point. Finite rational individuals recognize a difference between subjective and objective value, such that we can be tempted to choose value for future purpose that prioritizes our subjective interest over the divine *telos*, which provides the objective standard by which all value is finally measured. In other words, it is available to us as finite rational individuals to choose a lesser value as our aim for the future, to prioritize self-interest (subjective value) over the comprehensive *telos* that reason implicitly recognizes as the objective standard of value. In contrast, Gamwell suggests in an email to the author from 4 January 2019, that "God is the one individual in which egoism and altruism necessarily coincide," such that God never confronts the conditions that make moral agency possible and moral choices necessary (the potential conflict between the lesser value of self-interest and the objective standard of value: the divine good).

entities), but this space is also sufficient to allow for significant natural and moral evil, in spite of God's universal influence to the contrary.

## 6. God Is Not a Moral Agent

One reason to favor the alternative account of divine perfection advanced here is that it avoids what I take to be an unforced error in these disputes. This involves treating God as a sort of super moral agent. As I have suggested, this follows unavoidably from the traditional understanding of God's causal relation to the world: as creating ex nihilo with the implication that whatever powers are found in the effect of God's act of creation exist preeminently in the divine cause. In this context, God's inability to do what a finite being can do reveals a deficiency in God because it is assumed that the powers of finite beings relate to God's powers as imperfect to perfect. Furthermore, as Sterba argues in his book, wherever we attempt to account for evil based on the limitations of finite moral agents—whether in terms of their willing or in their limited powers—substituting God's agency reveals the possibility of avoiding the evils in question. But this idea of "divine moral substitution" fundamentally misunderstands the metaphysical limitations that exist in terms of the real relations among actual entities—divine and non-divine—in the world. Again, God is not omnipotent. Finite individuals are hard facts of the world in relation to which God can exercise influence through the selection of natural laws and as understood by rational beings as our comprehensive *telos*, one the one hand, and in response to which God can act to maximize value for future purpose, on the other. God's greatness, however, does not imply that God's powers are substitutable for the powers of finite individuals; the relative powers of the two classes of actual entities (divine and non-divine) simply are not substitutable in this way. Following the neoclassical account, "greatness" means having whatever properties it is better to have than not to have, as compared to other conceivable individuals, and the powers of moral agency reflect a form of finite agency incompatible with the divine nature.

What are some of the ways in which we might characterize God's perfection, or greatness, with respect to the transcendental characteristics of existence exemplified by all actual entities, divine and non-divine? On the neoclassical account, God is the only individual whose existence is compatible with any state of affairs whatsoever. Finite individuals, in contrast, are incompatible with all sorts of conditions, rendering them existentially fragile in a way that God is not. Additionally, God is the only subject in direct relation with all other individuals (as a universal subject) and capable of exercising universal influence (both in terms of setting the governing laws of each cosmic epoch and as the comprehensive *telos* at which finite rational agency should aim in seeking to maximize value for future purpose). Each finite individual interacts with a very limited portion of the world, and its influence is circumscribed by its finitude in a way that God's is not. The way to put these points metaphysically is to argue that to exist is to be in relation; to be is to experience and be experienced. The existence of finite individuals is constrained by relationships compatible with their existence; God is strictly compatible with all possible relationships (excluded by no conceivable state of affairs, or non-competitive, existentially speaking), which is precisely what it means to describe God as eternal. Finite beings are related to some but not all existing states of affairs (imperfect relationality); God is related to all states of affairs (perfect relationality).

It is worth emphasizing here that this approach is metaphysically abstemious, which I take to be a virtue. This approach avoids the difficulties of something like Aquinas's "analogy of being," where there is a fundamental and insuperable difference between how the metaphysical categories apply to divine and non-divine individuals. On the traditional account, this divide ensures that we are always reasoning analogically when applying our categories of existence to God, and there remains a gulf in what we can infer from experience regarding the divine individual's nature. On the neoclassical account, there are no metaphysical exceptions. The transcendental conditions for reality as such apply to God and non-divine individuals without exception, while still permitting the distinction

between these existential categories. Perfect and imperfect can still operate here, since we can speak of God's existence as surpassing that of any finite individual's in the ways described above. Still, to exist in all cases is to exist in relation, and one difference between God and non-divine individuals is the scope of the relations and whether an individual's existence is competitive with others. As noted, where we relate partially (imperfectly) to the world and are fragile with respect to some states of affairs (vulnerable, finite), God relates to all of reality (perfect relationality) and is strictly compatible with all conceivable states of affairs (invulnerable, eternal).

If we understand power as a type of influence exercised through relationships, then God's power is unsurpassable by any finite individual, even if it no longer makes sense, strictly speaking, to characterize God as omnipotent. In this way, both God and finite beings have real power (the ability to influence states of affairs through relationships) as actual entities. It is the scope of God's relationality and influence that characterize divine perfection, not, for example, the ability to act locally as a finite individual to secure a particular outcome, where we might reasonably evaluate whether the choice made was value maximizing with respect to the available alternatives. By framing the difference between finite beings and God in terms of perfect and imperfect relationality, where power has to do with our ability to influence that to which we relate, we retain the ability to attribute perfection to God without committing the category mistake of attributing moral agency and moral goodness to God, properties properly associated with finite rational individuals whose actions always involve a choice between greater and lesser value for future purpose.

## 7. The Divine Good: Beyond Aristotle's First Mover

Divine agency necessarily maximizes value for future purpose, where that future is always God's own. This is achieved, in part, through God's universal influence on finite actual entities to contribute to the divine good. One way of understanding the nature of this influence is that God seeks the greatest unity in diversity (or creativity) possible as the object of divine experience, where God's choices for future purpose always maximizes this value in light of the available alternatives. Each actual entity is internally related to its past, so its richness of feeling depends on what is inherited from that past. God, in turn, is internally related to every actual entity, such that the richness of God's experience reflects the contributions of all to the divine good. Each actual entity decides for the future in light of its inheritance and the possibilities this inheritance permits with the aim of maximizing value for the future. This is how the many (the inherited past) become one (a single subjective experience by an actual entity) and are increased by one (as the choice for future purpose results in a novel datum of experience for other actual entities—including God—that exemplifies value to a greater or lesser degree).

As Franklin Gamwell suggests, if the good is a quality that is to be realized through activity, then goodness must characterize states of affairs as possible choices for future purpose. Finite rational agency involves a moral evaluation of possibilities for future purpose in terms of this characteristic, but all actual entities contribute value in light of their activities. For finite rational individuals, such decisions imply an all-things-considered evaluation, since any conceivable state of affairs can be contemplated as a possible choice of action for a rational will. The conclusion Gamwell reaches is that "only the character of all possible things can define the good—and moral teleology is defined by a comprehensive purpose whose telos is strictly metaphysical" (Gamwell 2020, p. 128). In addition, because this characteristic is used to evaluate choices among possible states of affairs, it must be something that different choices realize to different degrees, which is what makes choice among alternatives significant. "The good defined by the possible as such is a variable," Gamwell continues, "such that all actualities exemplify it, and all future possibilities if and when realized will or may exemplify it in greater or lesser measure . . . . The final real things exemplify 'the many become one, and are increased by one' (Whitehead 1978, p. 21), that is, exemplify creative unification for the future" (Gamwell 2020, p. 135). This unity

in diversity represents the metaphysical variable to be maximized, and God both chooses the natural laws for a particular cosmic epoch with this aim and serves as the ultimate recipient of the value realized through the exercise of real, non-trivial power on the part of finite actual entities.

And what of moral goodness? What distinguishes moral and non-moral goodness on this account is not the formal standard of value as such (unity in diversity, or creativity) but rather the distinctive contribution that moral agents make to the divine good through their freely made choices to contribute maximally to the divine purpose. On this account, moral goodness is a species of goodness more generally, a subset of the more general category of value to be maximized in the divine life. Understood this way, we should anticipate a comprehensive account of value as a category to which there can be both moral and non-moral contributions. This conforms to a standard distinction within ethics between moral and non-moral value. Understood this way, however, we immediately see that value must be defined in such a way that all contributions share a common form, even as members of the class can be differentiated between moral and non-moral with respect to how the contribution is made, where moral value is realized through the choice by moral agents between greater and lesser value for future purpose with respect to the divine good.

All value represents a contribution to the comprehensive unity in diversity (creativity) realized through divine activity (which is value maximizing by necessity). A mundane way that might help us to approximate this idea is the completion of a jigsaw puzzle. When we open a new puzzle and spread the pieces out on the table, we have an example of diversity (the variety of distinct pieces) but little unity—it is just a mess of individual bits that anticipate an integrated whole. Once completed, however, we find something interesting. The diversity is still present—all the pieces are still there—but now the pieces have been harmonized into a whole, a complex unity in diversity. We find pleasure and satisfaction in the resolution of that initial disharmony and diversity into this final, creative achievement, one in which the individual parts have not been lost or effaced but merely enhanced through their integration into a greater whole that is itself a new object of subjective experience. Consider now the totality of the cosmos, where the various pieces are not simply inert objects on which a single will operates but rather a collection of individuals in relation that respond to one another, always contributing finite value in the subjective experience of other individuals through their decisions about how to realize value for the future, all of which together become a single, comprehensive unity in diversity in the decisions that God makes for future value in the divine life.

So, what is the distinctive contribution that moral agents make to the divine good such that we need to distinguish between moral and non-moral value? God, as the cosmic individual with the capacity to influence universally, chooses the natural laws within which finite individuals act toward greater unity in diversity (value maximization). As the comprehensive *telos* that reason recognizes as a condition for the possibility of a rational choice among alternatives for future purpose, God lures rational individuals—those who act with self-understanding—to maximize value for God: the divine good. What makes the value of such choices "moral" as opposed to "non-moral" is that this capacity for acting with self-understanding includes the possibility of self-contradiction, the choice of purpose that contradicts reason's recognition of a comprehensive *telos* as our proper aim. This is the possibility of moral evil, the free choice of a self-understanding that denies the responsibility to maximize value, all things considered, where the ultimate standard is the comprehensive good realized in the life of God (the divine good).

Moral agents are special insofar as they have a capacity for choosing between good and evil, and moral goodness, formally speaking simply represents the exercise of finite freedom in an act of self-understanding that decides for this comprehensive *telos* as its proper aim. Moral evil, in contrast, involves the exercise of finite freedom in an act of

self-understanding that decides against this comprehensive *telos* as its proper aim.[4] God does not choose between good and evil, since God is necessarily value maximizing with regard to God's future: this is simply what it means for God to make a decision for future value in light of what God inherits from the past, which includes the decisions made by all other finite individuals for future value. As Gamwell puts it in an email to the author from 4 January 2019, "By relating internally to strictly all things in all of their detail, God's actualities must again and again decide to pursue maximal creativity in the future as such—precisely because the future as such is the future of God." God is the ultimate beneficiary of all value, including the value that results from the moral choices of finite rational beings; however, God is not a moral agent, and it would be a category error to include moral goodness among God's perfections, since God is not choosing among greater and lesser values in God's decisions for future purpose (that is, for or against the divine good as the comprehensive *telos*). We can and should distinguish between God's being value maximizing by necessity, on the one hand (the neoclassical account), and God's being a perfect moral agent who necessarily fulfills the moral law in God's choice among greater and lesser value, on the other (possessing a holy will, or moral perfection in the traditional sense).

Some additional clarification about God's activity might be in order here. While God's existence is necessary, God's actions have a contingent aspect. There may be options available to God that are equally value maximizing, either with respect to the choice of natural laws for a cosmic epoch or in response to the value God inherits from God's own past and from the contributions of non-divine actual entities. The choice among these options is contingent and non-moral, since any option chosen among this set would be value maximizing. God's activity, then, always satisfies the metaphysical conditions implied by God's nature (always value maximizing), but this does not mean that God's actions involve no actual choice among alternatives. After all, in the absence of alternatives from which to choose, no choice can be made. Only if we assume that there is always only one way to maximize value must we conclude that God's choices are necessary both in their formal (value maximizing) and substantive (the specific choice made among equally value-maximizing options) aspects. It is not obvious that value-maximizing choices always imply a single option, such that, in effect, God never chooses but merely acts in whatever way is necessary to maximize value for future purpose. In addition, the neoclassical account implies limits on God's foreknowledge (contrary to the classic account of divine omniscience) in light of the real freedom of finite actual entities within the limits of any cosmic epoch. In other words, God can anticipate how the ordering of a particular cosmic epoch will provide broad conditions for coordinating the activities of actual entities in the world, but God cannot know (because it is unknowable in principle) precisely how those actual entities will use their freedom under those conditions. It may be that God's value-maximizing choice involves uncertainties that preclude the resolution of conceivable alternatives to a single, necessary option. Thus, God must choose in light of those uncertainties, always, of course, with the aim of maximizing value for God's future experience.

While God's agency is not moral in the sense of involving the choice between greater and lesser value, God's existence is morally significant for us. This is the sense in which God stands in an asymmetrical relationship with moral value, benefitting from it but not producing it through divine activity. This is because the divine good is properly the rational *telos* of all our choices as finite rational individuals. This involves our understanding of the exercise of our finite agency as requiring a choice among alternatives for future purpose, where the rational standard is to maximize value, with the recognition that we can (and

---

4　This involves a self-contradiction, since such a choice simultaneously recognizes, at least implicitly, that every choice of self-understanding for the future involves the judgment that this rather than some available alternative is more valuable, all things considered, and only a comprehensive *telos* can provide a rational means of evaluating such choices, since it alone provides a comprehensive standard of value by which an all-things-considered judgment can be made. A choice for a lesser value (e.g., in preference of self-interest over the divine good), then, results in a contradiction, since it involves the simultaneous affirmation and denial of a comprehensive *telos* as the standard of objective value.

often do) choose lesser values. Still, it is never rational to choose a lesser value, all things considered, and reason affirms that it is the divine good that is the ultimate standard of value for us. Why is this so? As Gamwell suggests in an email from January 4, 2019, "Decision with understanding cannot relate to its own final nullity: such decision is 'all things considered,' and the thought that any value we achieve or difference we make will eventually be erased is meaningless. Unless there is something ultimate at stake in what we do, then ultimately there is nothing at stake." The future to which we make an ultimate difference is the future of God as the universal subject who exists eternally and who is internally related to all things in the world. The ultimate meaning and value of our actions, then, rests on the difference they make to the divine life, however else we might also value them. As an eternal subject always apprehending the whole, God's experience is the sole good to which our actions can contribute permanent value: the divine good.

On the neoclassical account, God is the only individual that exists necessarily because, as Gamwell indicates in an email to the author on 18 February 2021, "God is the one individual definable entirely in metaphysical terms." God includes both an absolute pole (God's abstract, eternal nature understood as non-competitive with all other states of affairs) and a relative pole (God as universal subject internally related to the world in all of its particularity). In terms of God's perfection, God alone both influences universally and is universally affected. God values all existing individuals with respect to their unique contributions to the divine life as part of the harmony (unity in diversity, or creativity) that God seeks through God's universal influence and that is realized in God's decisions about value for the future in which those contributions find their final significance. Again, in terms of God's perfection, this reveals how God's existence in relation is comprehensive (universal, perfect) in a way that meaningfully contrasts with our finite existence in relation (partial, imperfect).

## 8. God's Power and Moral Goodness

God's power should be understood in relation to what God contributes to the world, both through God's universal influence and as the *telos* toward which all finite individuals contribute value. Such power far exceeds the power of finite beings, though there are actions possible for finite beings that are not available to God. Again, having discarded creation ex nihilo, there is no reason to believe that divine and non-divine powers relate to one another as perfect to imperfect in the traditional sense. As I have noted (following Hartshorne), in terms of greatness, there are some powers that are better, all things considered, not to have, and God's greatness includes all of the power proper to God in light of God's unique metaphysical status. Finite rational individuals bear the ultimate responsibility for moral good and evil, and God's subjective experience is diminished by our moral failures (since our failures contribute less value than was possible had we chosen differently). Again, God does all that God can do to maximize value through God's universal influence (the choice of natural laws for a cosmic epoch) and as the sole universal subject in relation to which all value finds its ultimate reference (the comprehensive *telos* with respect to which finite rational individuals make moral choices), and this is enough to establish God's greatness in contrast to our limited influence and experience.

At issue, then, is not whether events and conditions in the world satisfy our expectations for God as a super moral agent (omnipotent, omniscient, and morally perfect) but rather what is proper to God given God's unique metaphysical status. That God's influence on the world is universal (all existing entities are influenced by the divine reality) is compatible with God's causal efficacy being limited locally in light of the actual freedom and power of contingent beings (the reality of non-trivial freedom and power in the existing members of the class of finite individuals acting under the laws of nature of a particular cosmic epoch). God provides structure and order, including a cosmic *telos*, but this power to shape the whole does not override the finite causal powers of actual entities, even as it exerts universal influence on their actions and lures rational individuals to maximize

value for God.[5] Here, the analogy of the conductor of an orchestra might offer some partial insights into divine activity, keeping in mind that the conductor, unlike God, is also a finite individual in this example.

There are things that the conductor can do in terms of ordering the actions of the individual players in an orchestra in ways that integrate their efforts into a harmonious whole. No one of the individual players can accomplish this, and in this sense, the conductor's power is unsurpassed by any of the other members of the orchestra. That being said, the conductor cannot prevent an individual performer from playing a sour note or missing her entrance, each of which mars the beauty of the whole production. A good conductor does all that a conductor can to encourage excellent musicianship, both in setting the conditions for performance generally and when engaged in conducting a particular performance. In both cases, the conductor exercises powers of influence and persuasion unavailable to the other members. If she is a good conductor, then she does all that is proper to her to promote musical excellence (value) and minimize disharmony (evil) as these relate to the musical performance of the orchestra through her unique influence on the other members. Still, the conductor does not play the instruments for the players, and the conductor's ultimate achievement involves her influence on, and response to, the decisions freely made by the individuals in the orchestra.

While the analogy is imperfect, since the conductor is herself a finite individual with the powers (and limits) appropriate to that status, it provides some insight into God's activity in relation to the world. God plays a cosmic role in harmonizing the activity of the members of the class of finite individuals to the degree possible given God's unique metaphysical status. God is the sole individual whose influence is felt by all of the members simultaneously (in the laws that structure a particular cosmic epoch), and it is God's experience alone that realizes the harmony of the whole that is possible in light of God's universal influence and the actual decisions made by finite individuals (the understanding of which provides the lure for rational individuals to choose the divine good as their comprehensive *telos*). For rational beings, the divine good provides the condition for the possibility of rational choices among possible options for future purpose, all things considered. We might understand evil (both natural and moral) as discordance within the harmony that God seeks to maximize through God's universal influence on the class of finite individuals. God cannot prevent all discordance as the result of local, non-divine activity, even as God does all that is within God's power to maximize value through the universal influence that God exercises (the natural laws that provide the conditions for coordinated activity among finite actual entities) and the choices for future purpose that integrate finite contributions into a cosmic whole, which also provides the comprehensive *telos* of finite rational individuals (moral agents). Thus, God's activity is value maximizing in the sense relevant to God's unique agency, even while it remains inappropriate to attribute moral perfection to God's nature, since, strictly speaking, God does not choose between better and worse alternatives for future value in the manner of moral agents.

### 9. A Role for Skeptical Theism

On this account, God is doing all that God can to prevent evil, and it is here that there might be an appropriate role for a version of skeptical theism: the argument that we cannot judge God's actions because we lack sufficient knowledge of the tradeoffs that God is making. This has to do with God's choice among possible options for cosmic order—the scheme of natural laws for a particular cosmic epoch. To be clear, however, this version of skeptical theism is very different from something like Michael Bergmann's. As Sterba presents it in chapter 5 of his book, Bergmann's version of skeptical theism is meant to operate under the rules of classical theism, where God remains omnipotent and morally perfect, in spite of the challenges presented by the problem of evil, and the insufficiency in

---

5　Whitehead writes: "More than two thousand years ago, the wisest of men [Plato] proclaimed that the divine persuasion is the foundation of the order of the world, but that it could only produce such a measure of harmony as amid brute forces it was possible to accomplish" (Whitehead [1933] 1961, p. 160).

our knowledge of the conditions under which God chooses are meant to insulate God from our moral judgments.

Something like what I have described as the "substitution hypothesis" is at work in that context, it seems to me, so that for any state of affairs in the world where we can imagine a standard moral agent failing to prevent some evil, God's agency could "substitute," thus preventing the evil in question. As a super moral agent (omnipotent), God possesses all conceivable power. Bergmann suggests that one way to preserve God's existence on the traditional account against the problem of evil is to argue that we lack all of the relevant knowledge necessary to evaluate God's particular moral choices in such contexts. Sterba argues, however, that there remain insuperable problems for this line of apology for divine inactivity.

My understanding of Sterba's argument is that to see the problem with skeptical theism's defense we do not need to focus on individual cases where we might remain uncertain as to whether God's failure to intervene might be evidence against God's power or goodness, where a particular tradeoff might potentially be justified had we all of the relevant information. Instead, we should consider more generally what moral goods God might be understood to be advancing and consider whether it is possible to attain those goods in a world with significantly less evil. I find myself persuaded that Sterba is right in this context; once we consider the range of goods that we might imagine God pursuing as an omnipotent moral agent, it is not beyond our capacity to judge whether the apparent tradeoffs evident in the world (e.g., permitting things like the Holocaust, the miseries of slavery, and the suffering of the innocent from accidents and disease) seem warranted. If we can conceive of alternative ways to order the world that achieves those goods while also avoiding horrendous evil, then the skeptical position is undermined. Framed this way, Sterba argues that we do have the relevant knowledge for making the kinds of moral judgments sufficient to sustain the objection from evil. There are conceivable worlds with less evil or possibility of evil in which a range of significant moral and non-moral goods can be realized in contrast to the arrangement of our actual world, and an omnipotent God should have chosen one of those alternatives.

My objection here, however, is that the exchange between Sterba and Bergmann presupposes the traditional account of omnipotence, omniscience, and moral perfection, where among the possibilities available to God are fundamental alterations in the powers of finite individuals. As the omnipotent creator, God can make whatever tweaks to the nature and powers of God's creatures, so it is always within God's power to realize any conceivable world, including worlds in which the power and freedom of finite beings are constrained in ways that allow for the various goods at which God might aim to be realized without the risk of horrendous moral outcomes that are all too evident in our actual world. On Sterba's account, take any good that might require the exercise of creaturely power and freedom, the world can be arranged structurally so that there is power and freedom sufficient for achieving these goods while also ensuring that misuse of power and freedom never produces horrendous evil. If such a world is conceivable, then the actual world presents an objection to the logic of the traditional God's existence.

In the case of the God of neoclassical theism, however, the skeptical argument operates a bit differently. God is not simply a super moral agent, perfect in will and omnipotent, in contrast to our conflicted wills and imperfect powers. God does not create finite individuals and bestow on them their particular natures and powers as imperfect exemplifications of divine qualities. Actual entities, both divine and non-divine, exemplify the transcendental characteristics of existence, though these characteristics are self-differentiating between the divine individual and the class of finite, continent individuals. On the neoclassical account, it is a category mistake to ascribe moral agency to God precisely because God does not act in the world under the same conditions as finite rational beings, conditions that permit choices between greater and lesser value for the future. In considering God's relationship to value, including moral value, we have to take God's unique metaphysical status into consideration, not as a super moral agent but as a necessary individual that influences and

experiences comprehensively in the manner previously discussed. Again, God is active in the world in two primary ways proper to God's metaphysical status.

At one level, God provides the fundamental laws and structures for a cosmic epoch within which relations among finite individuals play out. Such laws and structures play a significant role in coordinating the activity of finite actual entities and allowing for the emergence of various degrees of harmony and order (unity in diversity) through the exercise of their non-trivial freedom and power. This suggests the possibility of different systems of natural laws among which God might choose in establishing the structures within which finite individuals exercise their power and freedom and contribute to the divine good. Our ability to evaluate whether a particular natural system is better than some conceivable alternative almost certainly runs up against the skeptical objection that we lack sufficient perspective to judge God's choices at this level. This version of the skeptical argument does not run into the same objections raised by Sterba in the case of traditional theism precisely because we are no longer speculating about the tradeoffs being made in the world by a super moral agent where our moral knowledge and experience seem sufficient to render the relevant judgments. We are not in a position to make comprehensive judgments about the merits of different systems of natural laws with regard to the tradeoffs involved for maximizing value over the course of a cosmic epoch, where finite beings necessarily possess the non-trivial powers and freedom appropriate to their natures and to which God's response is always value maximizing.

At another level, God provides a comprehensive *telos* for the activity of finite rational beings. As the universal subject of experience, all value realized by contingent beings is value realized, ultimately, for God, including moral value. As finite beings come into existence and disappear, they contribute to a greater or lesser degree to the divine life (whatever else they can be said to accomplish). These contributions make an objective and eternal (though mostly trivial) difference to God. We might imagine, though, that the contributions of rational beings are significantly less trivial in their contributions as a result, in part, of our moral agency. On the whole, God influences both rational and non-rational individuals in ways that aim at greater harmony and less discordance through the natural laws selected for a cosmic epoch; however, the power of actual entities is real and non-trivial, and the freedom of moral agents includes the choice between greater and lesser value for future purpose. The value available to be maximized by God's choice for the future is limited by the free choices of finite individuals, but the potential disharmony is also minimized as far as God's universal influence coordinates activity in the direction of greater creativity and moral choices are made by finite rational individuals in light of their understanding of the divine good as the comprehensive *telos*.

There are things we can do locally that God cannot, e.g., save a child from drowning, but examples like this simply reveal a difference between God's agency and the agency of finite individuals, not evidence of divine weakness or imperfection. Only if we start with traditional accounts of omnipotence would God's inability to act in the manner of a finite individual imply a deficiency in God, since on that account, any power found in a creature must be found more perfectly in God as creator. In contrast, the God of neoclassical theism offers an alternative understanding of divine perfection (God's greatness) that does not run afoul of the standard form of the objection from evil. God's activity is necessarily value maximizing in response to the acts of finite individuals who exercise non-trivial freedom in their own choices for future value. A God so understood, I believe, can serve as the proper object of our ultimate concern as finite rational beings even if this God is not the God of traditional theism in the Abrahamic traditions.

## 10. Another God of the Gaps?

There is a final area of concern might be helpful to address. In my original correspondence with Sterba, the issue arose as to whether the conception of deity I was defending resulted in "one cause too many" when discussing God's relationship to the world. I believe that Sterba was concerned that the neoclassical deity who operates comprehensively

and intimately in terms of being related to all existing things and exercising universal influence generates a "God-of-the-gaps" problem with respect to the inquiries of the natural sciences. His suggestion was that we did not need a God so understood to explain the cosmos, favoring instead the empirical findings of the natural sciences. It might be helpful to spend a little time on this topic, since it is a perennial one in philosophical theology and cosmology.

As I suggested previously, God exerts cosmic influence, but our knowledge of divine activity is not empirical. This knowledge is properly transcendental, the result of reflection about the nature of reality as such in light of common human experience and reason, and it is not the product of the observation and measure of particular, contingent events in the world. God sets the general conditions for a cosmic epoch within which finite individuals exercise their freedom and powers as actual entities, and God provides a comprehensive *telos* that serves as the condition of the possibility of making all-things-considered judgments of value for future purpose by finite rational individuals (moral agents). It is with respect to the divine experience of the whole that all value makes its ultimate contribution. If reason commands choices that maximize value for future purpose, then God's future is the only purpose that gives such choices ultimate significance, since only in God are the differences such choices make preserved for eternity (and a choice that makes no ultimate difference for the future is ultimately meaningless). Neither of these divine activities—establishing the laws of a particular cosmic epoch or serving as our comprehensive *telos*—are within the purview of the natural sciences to investigate.

The natural sciences are very helpful for describing the furniture of the cosmos and the various ways in which that furniture is arranged, including the natural laws that govern a particular cosmic epoch. But the natural sciences do not take up fundamental philosophical questions regarding how the actual conditions found in the universe relate to the range of alternatives that might be possible (e.g., why these particular cosmic constants—natural laws—as opposed to some conceivable alternatives?). The natural sciences do not consider whether the transcendental conditions of possible existence require that "something exists" is necessary or whether "God exists necessarily." These simply are not the kinds of topics addressed by the natural sciences, and the philosophical inquiry into these matters does not directly impinge on the empirical methods of those disciplines.

Similarly, the natural sciences are methodologically agnostic about whether there is any purpose in the universe, including anything like a comprehensive *telos* in relation to which determinations of value are ultimately made by rational beings such as ourselves. The account of value and the understanding of moral agency previously provided suggest that what is distinctive about our activity as finite rational beings is that it involves a choice among possible alternatives for the future, where reason directs us to maximize value but where the choice of lesser value is always available to us. The choice among values must make a difference for the future (otherwise, the choice is ultimately meaningless), and the denial that there is anything in terms of which different choices can be rationally evaluated with respect to the future nullifies the possibility of rational choice; such a denial is self-defeating. I have suggested that the neoclassical God provides the necessary *telos* with respect to which such choices can be rationally made. Only by contributing to the divine good do the efforts of finite beings make a permanent difference for the future, and the value of those contributions is ultimately measured by their positive contribution to the divine life. The divine good is the condition of the possibility of the meaningful choice of purpose for finite rational beings. Again, the natural sciences simply have nothing to say about a cosmic *telos* of this sort or the role it plays as a transcendental presupposition of our practical reasoning (as a condition for the possibility of rational choice among options for future purpose).

There is another point of distinction that I also think is worth making here. Regarding the role of something like astrophysics as a mode of empirical enquiry, I have every confidence in its ability to contribute to our knowledge of the contingent features of reality as they comprise an object of knowledge suitable to the methods of that discipline

(including the actual laws of nature for a particular cosmic epoch). That said, there is no discipline within the empirical sciences—astrophysics included—that can provide the basis for an experience of the universe as a whole. By "experience," I literally mean that some subject is capable of experiencing the referent of a concept, not merely that a subject is warranted in the use of the concept as a meaningful abstraction. We can have a concept of the universe as a whole, but it is not an object of experience for us and can never be, given our finitude. This is what makes the concept of the universe as a whole an abstraction for us: it is a concept that may have an objective referent, but that cannot be confirmed by our experience. In principle, "reality as a whole" is always merely an abstract idea for finite rational beings. Another way to put this point is that the abstract becomes concrete in experience, and for the cosmic whole to be more than an abstract idea, this whole must be an object of actual experience for some subject.

This inability to comprehend the whole in our experience is not merely a question of available technology or the need for innovations in our methods. As finite parts of the whole to which the concept of the whole refers, we simply are incapable of such an experience in principle. As finite members of this whole, the totality cannot be an object of discrete experience for us, and yet we cannot avoid presupposing that the whole exists as a concrete totality in spite of our fragmentary experiences of its various parts. It is a methodological presupposition of empirical cosmology. What grounds such a presupposition? If the abstract becomes concrete in experience, then for the cosmos to exist as a single, integrated whole, not merely as an abstract idea but in concrete specificity, then this implies some meaningful way in which that whole is an object of experience. To exist as something concrete is to be experienced in concrete specificity by a subject. To my mind, the only candidate for such an experience of the whole universe as a concrete totality is God.

The divine experience renders the cosmos a unity in diversity in concrete specificity, not merely as an abstract inference from finite experience but as an actual object of divine experience. Here, the empirical sciences are rendered moot, since there is no way, in principle, for the natural sciences to provide anything more than an abstract conception of the whole as a methodological presupposition for ongoing empirical inquiry. All of this is simply to suggest that nothing in the neoclassical account implies a conflict or competition with the methods and findings of the natural sciences (astrophysics included). The existence and activity of the neoclassical God does not attempt to provide supernatural solutions for natural mysteries that we can anticipate being resolved at a later time through innovations in the technologies and methods of the empirical sciences. Instead, this understanding of God does conceptual work for us in accounting for the initial choice of cosmic constants (the natural laws of a cosmic epoch selected by God), providing the ultimate grounds of practical reason for finite individuals (the divine good), and grounding the methodological presupposition of the natural sciences that our abstract conception of the universe exists as a concrete, unified whole (as an abstraction made concrete in the subjective experience of God). This is no God of the gaps.

## 11. Conclusions

As I indicated at the outset, my efforts in this essay were meant to be suggestive. I have not sought to provide systematic presentation of neoclassical theism in complete detail. My goal has been to challenge the idea that the problem of evil for traditional theism constitutes an objection to the logic of theism generally. As a result, I did not seek to counter Sterba's arguments against traditional theism and its apologists within this dispute. Instead, I have tried to argue that the conclusion we should reach as a result of his efforts is not that the existence of God is logically impossible but rather that we need to rethink our understanding of God's nature to avoid mischaracterizing God's existence as vulnerable to the problem of evil in the first place. To that end, I have tried to highlight what a neoclassical conception of God might do for us, with particular attention to how such an

account overcomes various objections to which traditional theism appears vulnerable with respect to the problem of evil.

In my original correspondence with Sterba, there seemed to me to be two main objections that are related to one another. I believe that at least part of his response was based on my initial mischaracterization of God as morally perfect. I am grateful to Gamwell for helping me to see my error in continuing to attribute moral perfection to God, as if God were simply a special type of moral agent. In that original context, Sterba's objection that the neoclassical God is an extremely weak moral agent made sense. This had to do with my insistence that while God affects the general conditions under which moral value is realized, God is incapable of acting locally to prevent moral evil. I believe I have addressed that error in this essay. While it may be counterintuitive in the context of traditional theism, the solution is to reject both omnipotence and moral perfection as divine attributes. To that end, I have endeavored to clarify both what this would mean in a neoclassical context and how the result is still a compelling account of God's nature that distinguishes God's greatness in contrast to our finite limitations. I hope I have done a better job here showing that moral agency applies properly to a subset of finite individuals—those with the capacity to choose between greater and lesser value for the future—and represents a category mistake when applied to God, whose actions are, by necessity, always value maximizing in relation to the whole. It is simply the nature of God's activity to maximize value for future purpose, where that future purpose, all things considered, is God's subjective experience of the cosmic whole.

The second objection that seemed particularly important to me from that original correspondence had to do with the problem of God's abstractness as an eternal being. The suggestion was that the eternal nature of the neoclassical God, like Aristotle's First Mover or the God of deism, renders God infinitely remote from ongoing events in the world, shielded from the problem of evil but also of little import to the moral lives of finite individuals. I have tried my best to address that concern by drawing on the process distinction between the eternal and consequent aspects of God's nature. God is existentially non-competitive and invulnerable (eternal), on the one hand, and God is affected by all other individuals as a universal subject (God's consequent nature), on the other. God is an eternal subject in intimate relationship with the world and provides the comprehensive *telos* in reference to which all value (moral and non-moral) finds its ultimate significance. In doing so, I have tried to steer between the danger of describing God's agency in a way that would inadvertently result in God's falling into the category of moral agents (and to whom the property of moral perfection would then apply and for whom the problem of evil would loom large), on the one hand, and leaving God's relationship to the world so obscure as to render God's existence practically meaningless (like Aristotle's First Mover or the deists' God), on the other. The reader can judge whether I have enjoyed any success in this effort.

The neoclassical tradition is rich and varied and represents a distinct alternative to classical theism. I believe that one of its major virtues is that it avoids the characterization of God's nature in a manner that is vulnerable to the problem of evil while still providing us with a robust framework for philosophical theology. I want to thank Jim for the opportunity to explore these issues further and for his assistance in thinking through these matters more systematically (both through our correspondence and in his fine book). I would also like to express my profound gratitude to Franklin Gamwell for his contributions to my thinking on these issues. Whatever clarity I bring to these matters is largely the result of his guidance. The deficiencies that remain are wholly my own.

**Funding:** This research received no external funding.

**Data Availability Statement:** Not applicable.

**Conflicts of Interest:** The author declares no conflict of interest.

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
