# Peer review of "God and the Problem of Evil: An Attempt at Reframing the Debate"

_religions, doi:10.3390/rel12030218_

Round 1

Reviewer 1 Report

I would recommend getting rid of the chattiness of introduction and I never really got a good sense of what value maximizing actually is -- is it simply increasing unity-in-diversity? -- and how that is not then also, in fact, moral? In addition, I was never sure what kind of actions this value-maximizing God was supposed to be doing and how those actions made contact or not with the storied concreteness of God and God's people. Is it value maximizing to resist the Roman occupation, and so on. If it is, then how is that not also moral. If it isn't, then how is that not also not-moral? In other words, I'm not convinced that "value maximizing" actually gets you out of any trouble as I am not sure that you can maintain the distinction when you really need it. And, if you do maintain that distinction, I'm not sure what 'God' you are talking about...maybe just the idea of God, as such, which isn't particularly interesting or helpful.

Author Response

Many thanks to reviewer 1 for the feedback! I've made revisions to the manuscript that I hope are sufficient. Please see the attachment for further details.

Reviewer 2 Report

The article is interesting, informative, and well written.  I have a couple of comments.

The author writes that “God does all that God can do to maximize value through God’s universal influence and as the sole universal subject in relation to which all value finds its ultimate reference, and this is enough to establish God’s perfection in contrast to our limited influence and experience.” (475-6). “God is doing all that God can to prevent evil” (527).

But this is precisely Sterba’s point:  God does not appear to be maximizing value within the scope of his power.  Granted that God as “unsurpassable” in power does not give him power to do just anything, but it seems that he should be able to do more, to be more persuasive, even given the inherent freedom of other actual entities.  In fact, Sterba’s point is that the freedom of some can legitimately be restricted if such promotes greater overall freedom (or value).  Sterba rejects the contention that the freedom of beings (actual entities) is completely sacrosanct. Would not God’s overall value be enhanced if there was less evil and more individuals contributing to God’s value?   This might involve some coercion in addition to persuasion, but so what if it brings about a greater value and less evil, contributing the God’s value.  (See the two articles on Process perspectives on evil in Michael Peterson, The Problem of Evil, 2nd edition, Notre Dame, 2017, 288-326.)

Author’s reply:  this objection presupposes the traditional view of God’s omnipotence, where God can override individual freedom, that is, act by substitution (301).  Rather, God’s actions have to do with choosing among and setting “the possible options for cosmic order – the scheme of natural laws” (529-30).  God is necessarily doing all that God can do, but his acts, like his existence, are not contingent, and are more general, relating to the cosmic order, than affecting particular individuals.

Sterba’s reply, I take it, is that this leaves God very distant from other actual entities.  His telos is our telos, but though God is related to us, the relationship is asymmetrical.  God is compatible with everything else (325), but takes no local action with other individuals.  That is, God has no meaningful interaction in the meantime (764f) other than providing “the comprehensive telos” (773).  God apparently does not act locally (340), is uninvolved in particularities, only the general telos. 

The author is unclear here.  On the one hand the author writes that “God is incapable of acting locally to prevent moral evil” (752); God only affects the “general conditions”, he does not “act locally to secure a particular outcome” (351).  God “does not choose between better and worse alternatives for future value” (524).   On the other hand, the author claims that God has “real power, the ability to influence states of affairs through relationships” (348).  He exercises influence (277), “lures individuals” (400).  Clearing this up would help the discussion.

Finally, the author claims that God makes decisions (238).  God chooses and provides natural laws (366, 590, 595).  God makes decisions about how to realize future value in God’s life (399).  “God seeks the greatest unity-in-diversity (or creativity) possible through harmonizing the decisions of finite individuals into a cosmic whole” (364).  The author’s use of words in reference to God, like “decides,” “chooses” (595) and “seeks” (364), seems out of place with his view of God who acts necessarily and is not a free, deliberative agent.  Since some natural laws are better than others, would not God choose the best options, i.e., those that better maximize value?  This language of choosing, deciding, and seeking, even in a cosmic sense, reflects an intentional agent involved in selecting between options, based on some principle of best. 

340  it should be “a” difference, not “the” difference.  The author notes other differences.

527 “It is here that here”.    Just “here” is needed

734 “systemic” should be “systematic”

  1. “Got” should be “God”

Author Response

Many thanks to reviewer 2 for the thoughtful feedback! I hope my response and revisions address his or her concerns. Please see the attachment for details.

Round 2

Reviewer 1 Report

I think the revisions are fine. I take your point that the discussion is metaphysical, but as a reader I would say that if your argument is successful, then you have thought of a God -- a transcendent principle -- who may be there but isn't worth talking about because it actually makes little difference one way or another. This is despite your claim that God's existence is morally significant for us and that God is an agent, though not a moral one. If it is somehow relevant, theoretically, and you maintain that we can meaningfully talk about divine activity then it doesn't seem all that different, subjectively, from someone who says "well, God must have a plan even though I don't understand what it is." The subjective moral evaluation of whatever is happening is undercut and drained from the event by metaphysical and epistemological distance. I guess it is nice to think that everything will tend towards unity-in-diversity under the divine influence, all things being equal, but as I said I'm not sure it all that helpful.

I agree with you, by the way, that God and matter are co-eternal and I also agree with you that the way the relationship between God and world is set up in orthodox theology is the source of the problem of evil. I disagree that the answer is to remove moral agency from God (for the reasons above). [Christian Early, “Divine Action in a Dynamic World: Towards a Peaceful Understanding of Active Matter and a God of Love,” in Practicing to Aim at Truth: Theological Engagements in Honor of Nancey Murphy (Eugene, OR: Wipf and Stock).]

My disagreements aside, however, the discussion about what kind of being God is and what can and cannot be said about God is significant...and in that discussion your proposal deserves to be heard and engaged with critically. Thus I am recommending the publication of the article.